

# IMEX_SfloW2D 1.0. A depth-averaged numerical flow model for pyroclastic avalanches

Mattia de' Michieli Vitturi[1], Tomaso Esposti Ongaro[1], Giacomo Lari[2], and Alvaro Aravena[3]

[1]Istituto Nazionale di Geofisica e Vulcanologia, Sezione di Pisa, Via della Faggiola 32, 56126 Pisa, Italy
[2]Università degli Studi di Pisa, Dipartimento di Matematica, Largo Pontecorvo 1, 56126 Pisa, Italy
[3]Università degli Studi di Firenze, Dipartimento di Scienze della Terra, Via La Pira, 50121 Firenze, Italy

**Correspondence:** Mattia de' Michieli Vitturi (mattia.demichielivitturi@ingv.it)

**Abstract.** Pyroclastic avalanches are a type of granular flow generated at active volcanoes by different mechanisms, including the collapse of steep pyroclastic deposits (e.g., scoria and ash cones) and fountaining during moderately explosive eruptions. They represent end-members of gravity-driven pyroclastic flows, characterized by relatively small volumes (less than about 1 Mm$^3$) and relatively thin (1-10 m) layers at high particle concentration (1-50 vol.%), manifesting strong topographic control. The simulation of their dynamics and mapping of their hazards pose several different problems to researchers and practitioners, mostly due to the complex and still poorly understood rheology of the polydisperse granular mixture, and to the interaction with the complex natural three-dimensional topography, which often causes rapid rheological changes. In this paper, we present IMEX_SfloW2D, a depth-averaged flow model describing the granular mixture as a single-phase granular fluid. The model is formulated in absolute Cartesian coordinates (where the fluid flow equations are integrated along the direction of gravity) and can be solved over a topography described by a Digital Elevation Model. The numerical discretization and solution algorithms are formulated to allow a robust description of wet-dry conditions (thus allowing to accurately track the front propagation) and to implicitly solve the non-linear friction terms. Owing to these features, the model is able to reproduce steady solutions, such as the triggering and stopping phases of the flow, without the need of empirical conditions. Benchmark cases are discussed to verify the numerical code implementation and to demonstrate the main features of the new model. A preliminary application to the simulation of the February 11th pyroclastic avalanche at Etna volcano (Italy) is finally presented. In the present formulation, a simple semi-empirical friction model (Voellmy-Salm rheology) is implemented. However, the modular structure of the code facilitates the implementation of more specific and calibrated rheological models for pyroclastic avalanches.

## 1 Introduction

Pyroclastic avalanches are rapid flows of pyroclastic material (volcanic ash, lapilli, pumices and scorias) that propagate down the volcanic slopes under the effect of gravity. They share many phenomenological features with other natural granular avalanches (such as landslides, debris flows, and rocks and snow avalanches) and they pose similar modelling, monitoring and risk mitigation challenges (Pudasaini and Hutter, 2007).

Despite the term pyroclastic avalanche is not widely used among volcanologists, who often adopt the term pyroclastic density current (PDC), there are reasons to prefer the former in some specific cases. The term avalanche derives from the french *avaler*,



i.e. literaly, "move down the valley". It is implicit the idea that a pyroclastic avalanche must move along and be confined within the volcanic slopes and be dominantly driven by the longitudinal (i.e., parallel to the ground) component of gravity. Pyroclastic avalanches stop when either the slope reduces or their momentum is dissipated by friction. This is opposite to a pyroclastic density current, a term deriving from the general concept of density currents (Von Karman, 1940; Benjamin, 1968; Simpson, 1999). Pyroclastic density currents are able to propagate far from the volcano even on flat topographies and move under the dominant action of the hydrostatic pressure associated with their large thickness and their density contrast with respect to the atmosphere (Esposti Ongaro et al., 2016). Friction is generally negligible in PDCs, which generally stop because of the progressive increase of buoyancy due to the combined effect of air entrainment and particle sedimentation (Bursik and Woods, 1996), which may give rise to the formation of co-ignimbrite eruption columns (Woods and Wohletz, 1991; Engwell et al., 2016), or because they cannot overcome topographic obstacles (Woods et al., 1998). Low aspect ratio ignimbrites (Fisher et al., 1993; Dade and Huppert, 1996; Dade, 2003) or flows produced by the collapse of Plinian columns (e.g., Shea et al., 2011) can generally be described as inertial PDCs for most of their runout. The two phenomenologies often overlap and, in some cases, pyroclastic avalanches show unexpected flow transformations, marking a transition to an inertial behavior, in which flow mobility is drastically enhanced (Fisher, 1995; Komorowski et al., 2015).

We will refer to pyroclastic avalanches in this work for those flows that 1) remain confined within the volcano slopes; 2) show evidence of a dense basal granular flow; 3) are controlled by topography (i.e., they mostly move in the direction of the maximum slope). Such conditions generally reduce the applicability of this cathegory to relatively small flows (less than about 1 million of cubic metres) generated by mildly explosive activity (e.g., Strombolian) or by the gravitational collapse of basaltic scoria cones or of relatively small viscous and degassed lava domes. It is worth remarking that this is also the volume threshold identified by Ogburn and Calder (2017) above which modelling of pyroclastic currents becomes more problematic, showing transitional features between the two phenomenologies.

## 1.1 Modelling and numerical simulation of shallow pyroclastic avalanches

As in classical fluid dynamics, even more so for granular fluids, the choice is between continuum and discrete field representation (Guo and Curtis, 2015). In this work, we prefer the continuum approach, which is more suited to large geophysical systems (that would otherwise require a prohibitive number of discrete particles). As in similar models already considered in volcanological research and applications (Pitman et al., 2003; Kelfoun and Druitt, 2005; Shimizu et al., 2017), we also adopt here a physical formulation based on depth-averaging, which is appropriate for shallow granular avalanches and it is computationally less expensive. Finally, the model is formulated for a single granular fluid. Future developments and implementations will consider multiphase flows as a more accurate representation of pyroclastic avalanches (Dufek, 2015).

Despite these simplifying hypotheses, several difficulties arise in granular avalanche depth-averaged models. On one hand, terrain-following coordinates are often used to express the depth-averaged transport equations. However, on 3D rough surfaces, they need to be corrected with curvature terms, which introduce problems with irregular topographies, cliffs, obstacles and high curvatures (Denlinger and Iverson, 2004; Fischer et al., 2012). Moreover, on steep slopes, where acceleration along $\hat{z}$ is non-negligible, the hydrostatic approximation is flawed (Denlinger and Iverson, 2004; Castro-Orgaz et al., 2015; Yuan et al., 2017).





On the other hand, from a physical point of view, the description of the depth-averaged rheology of the granular fluid revealed to be problematic for strongly stratified and non-homogeneous flows (Bartelt et al., 2016; Kelfoun, 2017; Shimizu et al., 2017). Even in the more recent literature, a unifying model for the rheology of fast granular flows is still lacking (Bartelt et al., 1999; Iverson and Denlinger, 2001; Mangeney et al., 2007; Forterre and Pouliquen, 2008; Kelfoun, 2011; Iverson and George, 2014; Lucas et al., 2014; Delannay et al., 2017).

In addition to the latters, some additional difficulties arise from the numerical solution of the conservation equations: despite numerical method based on conserv ative, approximate Riemann solvers are robust and well tested (Denlinger and Iverson, 2001; Mangeney-Castelnau et al., 2003; Patra et al., 2005; Christen et al., 2010; Toro, 2013), non-hydrostatic terms arising from the vertical momentum equation (Denlinger and Iverson, 2004) can be computationally expensive and/or need particular treatment. In many cases, further difficulties arise in the treatment of source terms (especially for thin flow, where friction dominates), often requiring empirical yielding/stopping criteria that might (and usually do) deteriorate the numerical solution (Charbonnier and Gertisser, 2009; Ogburn and Calder, 2017). Last, but not least, there are just a few open-source codes, and those available are not easy to modify due to the lack of documentation.

In this paper, we present and show verification tests of the new IMEX_SfloW2D numerical model for shallow granular avalanches, that we designed to address most of the above difficulties. In particular, the model is formulated in a geographical (absolute) coordinate system, so that it is possible to include non-hydrostatic terms arising from the steep topographic slopes or from more accurate approximation of the vertical momentum equations. The model can deal with different initial and boundary conditions, but its first aim is to treat gravitational flows over topographies described as Digital Elevation Models (DEMs) in the ESRI ascii format. The same format is used for the output of the model, so that it can be handled very easily with GIS software.

Numerically, the first and most relevant advancement is represented by the implicit treatment of the source terms in the transport equations, which avoids most problems related to the stopping of the flow, especially when dealing with strongly non-linear rheologies. The model is indeed discretized in time with an explicit-implicit Runge-Kutta method where the hyperbolic part and the source term associated with topography slope are solved explicitly, while other terms (friction) are treated implicitly. The finite volume solver for the hyperbolic part of the system is based on the Kurganov and Petrova 2007 semi-discrete central-upwind scheme and it is not tied to the knowledge of the eigenstructure of the system of equations. The implicit part is solved with a Newton-Raphson method where the elements of the Jacobian on the non-linear system are evaluated numerically with a complex step derivative technique. This automatic procedure allows to use different formulations of the friction term without the need of major modifications of the code. In particular, the Voellmy-Salm empirical model is implemented in the present version.

The FORTRAN90 code can be freely downloaded and it is designed in a way that the users can simply use it without any intervention or they can easily modify it adding new transport and/or constitutive equations.





## 2 Physical and Mathematical model

In this section we present the governing equations, based on the shallow water approximation. We omit their derivation, that comes from the manipulation of the mass conservation law and the Newton's second equation of motion.

### 2.1 Depth-averaged equations

The model we use for the fluid evolution is described by the Saint Venant's equations (Pudasaini and Hutter, 2007; Toro, 2013), coupled with source terms modeling frictional forces. Such partial differential equations assume that vertical flow motion can be neglected and model the changes in space and time of flow depth $h(x,y,t)$ and horizontal velocities $u(x,y,t)$ and $v(x,y,t)$ (averaged across the vertical column) over a topography $B(x,y)$ (we assume here that the topography does not change with time $t$). Here we write the equations in global Cartesian coordinates, thus the two velocities are defined as the components

along the $x$ and $y$ axes, orthogonal to the $z$ axis, which is parallel to the gravitational acceleration $\mathbf{g} = (0,0,g)$. In addition, in this work we assume an hydrostatic pressure distribution, resulting in the following relationship between pressure $p$, bulk density of the flow $\rho$ (assumed to be constant) and flow depth $h$:

$$p = \rho g h. \tag{1}$$

With these assumptions, and without considering frictional forces, the 2D inviscid depth-averaged equations in differential

form can be written in the following way:

$$\frac{\partial h}{\partial t} + \frac{\partial (hu)}{\partial x} + \frac{\partial (hv)}{\partial y} = 0, \tag{2}$$

$$\frac{\partial (hu)}{\partial t} + \frac{\partial \left(hu^2\right)}{\partial x} + \frac{\partial (huv)}{\partial y} + gh\frac{\partial (h+B)}{\partial x} = 0, \tag{3}$$

$$\frac{\partial (hv)}{\partial t} + \frac{\partial (huv)}{\partial x} + \frac{\partial \left(hv^2\right)}{\partial y} + gh\frac{\partial (h+B)}{\partial y} = 0. \tag{4}$$

The first equation represents conservation of mass, while the other two equations describe conservation of momentum in $x$

and $y$ directions. The last terms in Eqs. (3-4) account for the gravity-induced force, which results from the hydrostatic pressure and depend also on the bathymetry $B(x,y)$.

As stated above, the variables of the model are $(h,u,v)$, but to deal more easily with the topography we introduce the additional variable $w = h + B$, describing the height of the free surface of the flow. Please note that if we substitute $w$ to $h$ in the temporal derivative of Eq. (2), this still represents mass conservation, because we assume $B$ not changing with time.

If we introduce now the vector of *conservative variables* $\mathbf{Q} = (w, hu, hv)^T$ (where the superscript notation $\mathbf{Q^{(i)}}$ is used to denote the $i-$th component), the governing equations can be written in the compact form:

$$\mathbf{Q}_t + \mathbf{F(Q)}_x + \mathbf{G(Q)}_y = \mathbf{S}_1(\mathbf{Q}), \tag{5}$$





where the letter-type subscript denotes the partial derivative with respect to the correspondent variable, and the terms appearing in the equation are defined in the following way:

$$\mathbf{F}(\mathbf{Q}) = (hu, hu^2 + \frac{1}{2}gh^2, huv)^T,$$

$$\mathbf{G}(\mathbf{Q}) = (hv, huv, hv^2 + \frac{1}{2}gh^2)^T, \tag{6}$$

$$\mathbf{S}_1(\mathbf{Q}) = (0, ghB_x, ghB_y)^T.$$

We observe that the homogeneous part of Eq. (5) represents the shallow water equations over a flat topography, for which the eigenstructure and hyperbolicity are well studied (Toro, 2013). Furthermore, the homogeneous part is written in a conservative form, allowing to easily adopt finite-volumes discretization schemes for its numerical solution.

## 2.2 Voellmy-Salm rheology

To shallow pyroclastic avalanches, we have to modify the classic Saint Venant's equations introducing an additional source term $\mathbf{S}_2$ accounting for friction forces (Pudasaini and Hutter, 2007; Toro, 2013). To this purpose, we implemented in IMEX-SfloW2D, as a prototype non-linear model, the Voellmy-Salm rheology, which simulates the mixture motion as a homogeneous mass flow. This model is commonly used for avalanches and debris flows, but it fits also for volcanic gravitational flows.

The terms we consider appear only in the momentum equations ($S_2^{(1)} = 0$) and are given by:

$$S_2^{(2)}(\mathbf{Q}) = \frac{u}{\sqrt{u^2 + v^2}}\left[\mu h\mathbf{g}\cdot\mathbf{n} + \frac{g}{\xi}(u^2 + v^2)\right],$$

$$\tag{7}$$

$$S_2^{(3)}(\mathbf{Q}) = \frac{v}{\sqrt{u^2 + v^2}}\left[\mu h\mathbf{g}\cdot\mathbf{n} + \frac{g}{\xi}(u^2 + v^2)\right].$$

The total basal friction in the Voellmy-Salm model (represented by the common term in square brackets in the two right-hand sides of Eqs. (7)) is split into two components: 1) a velocity independent dry Coulomb friction, proportional to the coefficient

$\mu$, the flow thickness and the component of the gravitational acceleration normal to the topography; and 2) a velocity dependent turbulent friction, inversely proportional to the coefficient $\xi$. For the sake of simplicity, $\mu$ is named the *friction coefficient* and $\xi$ the *turbulent coefficient*. If the topography is a function of the global coordinates $z = B(x, y)$, then the component of the gravitational acceleration normal to the topography is given by:

$$\mathbf{g}\cdot\mathbf{n} = \frac{g}{\sqrt{1 + B_x^2 + B_y^2}}. \tag{8}$$

With the notation introduced above, the final form of the equations modelling pyroclastic avalanches is:

$$\mathbf{Q}_t + \mathbf{F}(\mathbf{Q})_x + \mathbf{G}(\mathbf{Q})_y = \mathbf{S}_1(\mathbf{Q}) + \mathbf{S}_2(\mathbf{Q}), \tag{9}$$

where the hyperbolic terms and the source terms are defined by Eqs. (6-7). In this formulation we have kept the two source terms separated because, as shown in the next section, they require different treatments.



## 3 Numerical model

IMEX-SfloW2D is based on a finite-volume central-upwind scheme in space and on a implicit-explicit Runge-Kutta scheme for the discretization in time. The main purpose of the code is to run simulations on co-located grids derived from DEMs, and for this reason the standard input files defining the topography are raster files in the ESRI ASCII format, defining a uniform grid of equally sized square pixels whose values (in our case representing the terrain elevation above sea level) are arranged in rows and columns. The procedure to define the elevation values at the face centers and cell centers of the computational grid is represented in Fig. 1. First, the pixel values of the ESRI file (represented with colored squares in Fig. 1) are associated to the coordinates of the center of the pixels (filled circles in Fig. 1), and then these values are linearly interpolated at the four corners of the computational grid (no-fill circles in Fig. 1). Finally, the elevation values $B_{j,k}$ at the centers of each cell (filled squares in Fig. 1) are defined as the average value of the four cell corners, while the values at the centers of each face, denoted with $B_{j,k+\frac{1}{2}}$ and $B_{j+\frac{1}{2},k}$ and represented by the no-fill squares in Figure 1, are defined as the average value of the two face corners. With this definition, $B_{j,k}$ will also be the average of the values at the centers of the four faces of the cell $(i,j)$ and this fact plays an important role for a correct numerical discretization of the last two terms in Eqs. (3-4), resulting in a scheme capable to preserve steady states.

### 3.1 Central-Upwind scheme

The finite-volume method here adopted is based on the semi-discrete central-upwind scheme introduced in Kurganov and Petrova (2007), where the term central refers to the fact that the numerical fluxes at each cell interface are based on an average of the fluxes at the two sides of the interface, while the term upwind is employed because, in the flux averaging, the weights depend on the local speeds of propagation at the interface.

Following Kurganov and Petrova (2007), the semi-discretization in space leads to the following ordinary differential equations system in each cell:

$$\frac{d}{dt}\overline{\mathbf{Q}}_{j,k}(t) = -\frac{\mathbf{H}^x_{j+\frac{1}{2},k}(t) - \mathbf{H}^x_{j-\frac{1}{2},k}(t)}{\Delta x} - \frac{\mathbf{H}^y_{j,k+\frac{1}{2}}(t) - \mathbf{H}^y_{j,k-\frac{1}{2}}(t)}{\Delta y} + \overline{\mathbf{S}}_{j,k}(t) \tag{10}$$

where $\overline{\mathbf{Q}}_{j,k}$ denotes the average of the conservative variables $\mathbf{Q}(x,y)$ over the control volume $(j,k)$ and $\mathbf{H}^x$ and $\mathbf{H}^y$ are the numerical fluxes, calculated from the value of the variables reconstructed at the cell interfaces.

The choice of the variables to reconstruct at the interface is fundamental for the stability of the numerical scheme. The homogeneous system associated to Eq. (9) admits smooth steady-state solutions, as well as non-smooth steady-state solutions. A good numerical method for the solution of the homogeneous system should accurately capture both the steady state solutions and their small perturbations (quasi-steady flows). From a practical point of view, one of the most important steady-state solutions is a stationary one:

$$w = h + B = const, \qquad u = v = 0. \tag{11}$$

This suggest to use, as the vector of variables for the linear reconstruction at the interfaces, the vector $\mathbf{U} = (w,u,v)^T$, denoted as the vector of *physical variables* of the system, for which the boundary conditions are also prescribed. IMEX_SfloW2D also





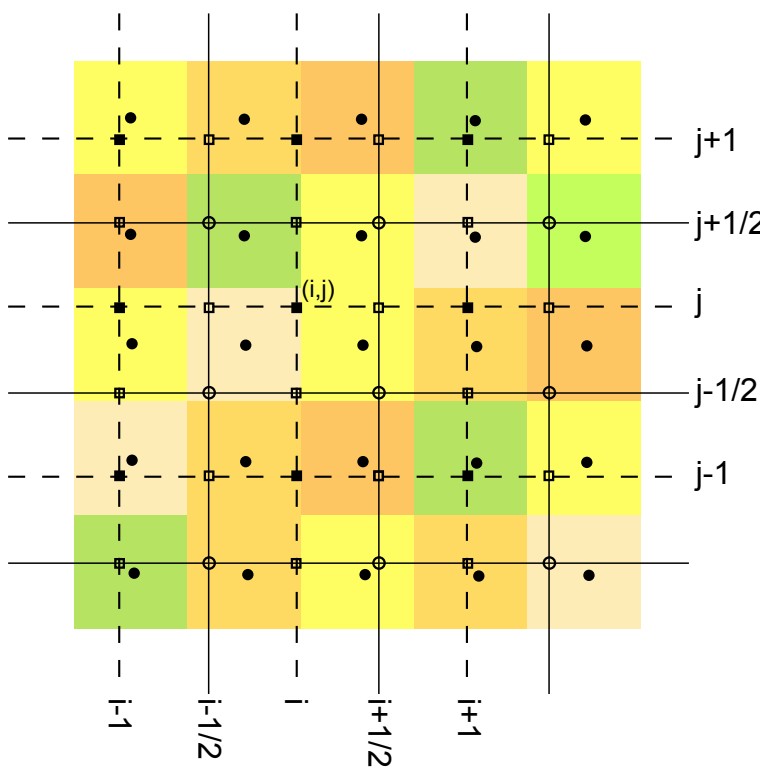

**Figure 1.** Computational grids. The colored pixels represent the elevation values of the original DEM. The lines define the edges of the IMEX-Sflow2D computational cells. The elevation values at the centers (filled squares), faces (no-fill squares) and corners (no-fill circles) of the computational cells are obtained by interpolating the pixel values, associated to their centers (filled circles).

allows the user to choose as set of variables for the linear reconstruction the vector of conservative variables $\mathbf{Q} = (w, hu, hv)^T$. In this case, a correction procedure is required to limit the values of the velocity components at the interfaces when the flow thickness goes to zero, as done in Kurganov and Petrova (2007).

For the reconstruction procedure based on the physical variables, we introduce the notation $\mathbf{\Gamma} : \mathbb{R}^3 \to \mathbb{R}^3$ for the mapping from conservative variables $\mathbf{Q}$ to physical variables $\mathbf{U}$, and $\mathbf{\Gamma}^{-1}$ for the inverse mapping from physical to conservative variables. From the average values of the physical variables we can operate a linear reconstruction inside each cell in order to obtain the values at the interfaces sides. In particular, given the local partial derivatives at the cell center $(\mathbf{U}_x)_{j,k}$ and $(\mathbf{U}_y)_{j,k}$, the one-side values at the East, West, North and South interfaces of the cell $(j, k)$ are given by:

$$\mathbf{U}_{j,k}^E = \overline{\mathbf{U}}_{j,k} + \frac{\Delta x}{2}(\mathbf{U}_x)_{j,k}, \qquad \mathbf{U}_{j,k}^W = \overline{\mathbf{U}}_{j,k} - \frac{\Delta x}{2}(\mathbf{U}_x)_{j,k},$$

$$\mathbf{U}_{j,k}^N = \overline{\mathbf{U}}_{j,k} + \frac{\Delta x}{2}(\mathbf{U}_y)_{j,k}, \qquad \mathbf{U}_{j,k}^S = \overline{\mathbf{U}}_{j,k} - \frac{\Delta x}{2}(\mathbf{U}_y)_{j,k}.$$





These partial derivatives are calculated using an opportune geometric limiter. In IMEX_SfloW2D it is possible to choose between MinMod, Superbee and Van Leer limiters. We observe that at each cell interface, for each variable, there are two reconstructed values, one from each cell at the two sides of the interface.

During the reconstruction step, a particular care should be taken in order to avoid unrealistic values of the physical variables, such as negative flow thickness or velocities too large. For this reason, in the case one of the reconstructed interface values of $w$ is smaller than the topography $B$ at the same location (thus resulting in a negative thickness $h$), the relative derivative is further limited to have a zero-thickness at such interface. We remark that the correction is applied to the derivative and thus also the reconstructed value of $w$ at the opposite interface will be affected. For example, if $w_{j,k}^S < B_{j,k-\frac{1}{2}}$, then we take the following derivative in the $y$-direction:

$$(w_y)_{j,k} = \frac{\bar{w}_{j,k} - B_{j,k-\frac{1}{2}}}{\Delta y/2}, \tag{12}$$

which gives the two reconstructions at the $S$ and $N$ interfaces of the $(i, j)$ control volume:

$$w_{j,k}^S = B_{j,k-\frac{1}{2}}, \qquad w_{j,k}^N = 2\bar{w}_{j,k} - B_{j,k-\frac{1}{2}}. \tag{13}$$

As stated above, an additional problem in the reconstruction step is that $h$ can be very small, or even zero. Thus, when the physical variables $u$ and $v$ at the interfaces are computed from the conservative variables, a desingularization is applied in order to avoid the division by very small numbers and the corrected values are given by the following formulas:

$$u = \frac{\sqrt{2}h(hu)}{\sqrt{h^4 + \max(h^4, \epsilon)}}, \qquad v = \frac{\sqrt{2}h(hv)}{\sqrt{h^4 + \max(h^4, \epsilon)}}, \tag{14}$$

where $\epsilon$ is a prescribed tolerance.

Finally, once the physical variables are reconstructed at the interfaces, the numerical fluxes in the $x$-direction are given by:

$$\mathbf{H}_{j+\frac{1}{2},k}^x = \frac{a_{j+\frac{1}{2},k}^+ \mathbf{F}(\mathbf{\Gamma}^{-1}(\mathbf{U}_{j,k}^E), B_{j+\frac{1}{2},k}) - a_{j+\frac{1}{2},k}^- \mathbf{F}(\mathbf{\Gamma}^{-1}(\mathbf{U}_{j+1,k}^W), B_{j+\frac{1}{2},k})}{a_{j+\frac{1}{2},k}^+ - a_{j+\frac{1}{2},k}^-}$$
$$+ \frac{a_{j+\frac{1}{2},k}^+ a_{j+\frac{1}{2},k}^-}{a_{j+\frac{1}{2},k}^+ - a_{j+\frac{1}{2},k}^-} \left( \mathbf{\Gamma}^{-1}(\mathbf{U}_{j+1,k}^W) - \mathbf{\Gamma}^{-1}(\mathbf{U}_{j,k}^E) \right) \tag{15}$$

where the right- and left-sided local speeds $a_{j+\frac{1}{2},k}^+$ and $a_{j+\frac{1}{2},k}^-$ are estimated by:

$$a_{j+\frac{1}{2},k}^+ = \max \left( u_{j,k}^E + \sqrt{gh_{j,k}^E}, u_{j+1,k}^W + \sqrt{gh_{j+1,k}^W}, 0 \right),$$
$$a_{j+\frac{1}{2},k}^- = \min \left( u_{j,k}^E - \sqrt{gh_{j,k}^E}, u_{j+1,k}^W - \sqrt{gh_{j+1,k}^W}, 0 \right).$$

In a similar way, the numerical fluxes in the $y$-direction are given by:

$$\mathbf{H}_{j,k+\frac{1}{2}}^y = \frac{b_{j,k+\frac{1}{2}}^+ \mathbf{F}(\mathbf{\Gamma}^{-1}(\mathbf{U}_{j,k}^N), B_{j,k+\frac{1}{2}}) - b_{j,k+\frac{1}{2}}^- \mathbf{F}(\mathbf{\Gamma}^{-1}(\mathbf{U}_{j+1,k}^S), B_{j,k+\frac{1}{2}})}{b_{j,k+\frac{1}{2}}^+ - b_{j,k+\frac{1}{2}}^-}$$
$$+ \frac{b_{j,k+\frac{1}{2}}^+ b_{j,k+\frac{1}{2}}^-}{b_{j,k+\frac{1}{2}}^+ - b_{j,k+\frac{1}{2}}^-} \left( \mathbf{\Gamma}^{-1}(\mathbf{U}_{j+1,k+1}^S) - \mathbf{\Gamma}^{-1}(\mathbf{U}_{j,k}^N) \right) \tag{16}$$



where local speeds in the $y$ direction $b^+_{j+\frac{1}{2},k}$ and $b^-_{j+\frac{1}{2},k}$ are given by:

$$b^+_{j,k+\frac{1}{2}} = \max\left(v^N_{j,k} + \sqrt{gh^N_{j,k}}, v^S_{j,k+1} + \sqrt{gh^S_{j,k+1}}, 0\right),$$

$$b^-_{j,k+\frac{1}{2}} = \min\left(v^N_{j,k} - \sqrt{gh^N_{j,k}}, v^S_{j,k+1} - \sqrt{gh^S_{j,k+1}}, 0\right).$$

Following Kurganov and Petrova (2007), the source term $\mathbf{S_1}$ is calculated trough a quadrature formula:

$$S_1^{(2)} = -g(w_{j,k} - B_{j,k})\frac{B_{j+\frac{1}{2},k} - B_{j-\frac{1}{2},k}}{\Delta x}, \tag{17}$$

$$S_1^{(3)} = -g(w_{j,k} - B_{j,k})\frac{B_{j,k+\frac{1}{2}} - B_{j,k-\frac{1}{2}}}{\Delta y}. \tag{18}$$

This discretization, coupled with the fact that the elevation $B_{j,k}$ at the center of the control volumes is defined as the average value of the elevation at the center of the faces, guarantees that the resulting second-order numerical scheme is well-balanced (i.e. preserves steady state solutions) and the solutions are non-negative.

## 3.2 Runge-Kutta method

The semi-discrete system of equations (10) is solved using an Implicit-Explicit (IMEX), Diagonally-Implicit Runge-Kutta scheme (DIRK), because such schemes are well-suited for solving stiff systems of partial differential equations, and the governing equations are expected to be stiff given the strong non-linearities present in the friction terms. In addition, an implicit treatment of these terms allows for a better coupling of the equations and to properly recover the stoppage condition without the need to impose additional criteria or arbitrary thresholds.

The family of IMEX methods (Ascher et al., 1997) have been developed to solve stiff systems of partial differential equations written in the form:

$$\mathbf{Q}_t + \mathbf{P}(\mathbf{Q}) = \mathbf{R}(\mathbf{Q}), \tag{19}$$

where in $\mathbf{P}$ are lumped all the non-stiff terms (in our case the semi-dicretized conservative fluxes $\mathbf{F}$ and $\mathbf{G}$ and the term $\mathbf{S_1}$), while $\mathbf{R}$ denotes the stiff terms of the system (here represented by the friction term $\mathbf{S_2}$). The system of equations (19) must be solved for each control volume $(j, k)$, but here, to keep the notation simpler, we omit the subscripts.

An IMEX Runge-Kutta with $\nu$ steps scheme consists of applying an implicit discretization to the stiff terms and an explicit one to the non-stiff terms, obtaining:

$$\mathbf{Q}^{(l)} = \mathbf{Q}^n - \Delta t \sum_{m=1}^{l-1} \tilde{a}_{lm}\mathbf{P}(\mathbf{Q}^{(m)}) + \Delta t \sum_{m=1}^{\nu} a_{lm}\mathbf{R}(\mathbf{Q}^{(m)}), \qquad l = 1,\dots,\nu \tag{20}$$

$$\mathbf{Q}^{n+1} = \mathbf{Q}^n - \Delta t \sum_{m=1}^{\nu} \tilde{b}_m\mathbf{P}(\mathbf{Q}^{(m)}) + \Delta t \sum_{m=1}^{\nu} b_m\mathbf{R}(\mathbf{Q}^{(m)}). \tag{21}$$





The choice of the number of the Runge-Kutta steps $\nu$, of the $\nu \times \nu$ matrices $\tilde{A} = (\tilde{a}_{lm})$ and $A = (a_{lm})$ and of the vectors $\tilde{b} = (\tilde{b}_1, \ldots, \tilde{b}_\nu)$ and $b = (b_1, \ldots, b_\nu)$ differentiates the various IMEX Runge-Kutta schemes. We remark that the explicit discretization of the non-stiff terms requires $\tilde{a}_{lm} = 0$ for $l \geq m$, while the implicit treatment of the stiff terms requires $a_{lm} \neq 0$ for some $l \geq m$.

Following Pareschi and Russo (2005), the IMEX scheme used in this work satisfy an additional condition, i.e. $a_{lm} = 0$ for $l > m$. This family of IMEX schemes are called Direct Implicit Runge-Kutta (DIRK) schemes and their use leads to the following implicit problem to solve at each step the Runge-Kutta procedure:

$$\mathbf{N}(\mathbf{Q}^{(l)}) \equiv \mathbf{Q}^{(l)} - \Delta t \cdot a_{ll}\mathbf{R}(\mathbf{Q}^{(l)}) - \mathbf{Q}^n + \Delta t \sum_{m=1}^{l-1} \left[ \tilde{a}_{lm}\mathbf{G}(\mathbf{Q}^{(m)}) + a_{lm}\mathbf{R}(\mathbf{Q}^{(m)}) \right] = 0. \tag{22}$$

To enforce the stopping condition that can result from the application of the total basal friction, the velocity independent friction
term and the velocity dependent term are computed in two steps. First, the dry Coulomb friction is computed and its value is limited to account for the fact that this force at maximum can stop the flow. Then, the system of nonlinear equations (22) in the unknowns $\mathbf{Q}^{(l)}$ is solved using a Newton-Raphson method with an optimum step size control. The method requires the computation of the Jacobian matrix $\mathbf{J}$ of the left-hand side of equation (22) with the highest possible accuracy, since $\mathbf{R}(\mathbf{Q}^{(j)})$, accounting for the dependence of the friction force on flow variables, can be strongly non-linear. Following Martins et al. (2003)
and La Spina and de' Michieli Vitturi (2012), this can be obtained with the use of complex variables to estimate derivatives. With the complex-step derivative approximation we can approximate the Jacobian $\mathbf{J}$ needed for the Newton-Raphson method with an error of the same order as the machine working precision. We simply extend the function $\mathbf{N}$ to the complex plane, introducing the new function: $\tilde{\mathbf{N}} : \mathbb{C}^3 \to \mathbb{C}^3$ and compute the real-valued columns of the Jacobian at $\mathbf{Q}$ as

$$\mathbf{J} \cdot \mathbf{e}_j = \frac{\Im(\tilde{\mathbf{N}}(\mathbf{Q} + i\epsilon\mathbf{e}_j))}{\epsilon} \tag{23}$$

where $(\mathbf{e}_j)_{j=1,\ldots,3}$ are the standard basis vectors of $\mathbb{R}^3$, $\Im(\cdot)$ denotes the imaginary part of complex numbers, and $\epsilon$ is a real number of the order of the machine working precision. Once the Jacobian is computed, the descent direction of the Newton-Raphson is updated and the descent step is obtained by applying a globally convergent method as described in Press et al. (1996).

## 4    Numerical tests and code verification

In this section we present numerical tests aimed at demonstrating the mathematical accuracy of the numerical model results (the *verification* step, following Oberkampf and Trucano, 2002). Numerical tests are aimed at proving:

1. the capability to manage the propagation of discontinuities;

2. the potential to deal with complex and steep topographies and dry/wet interfaces;

3. the ability of the granular avalanche to stop, achieving the expected steady state.



All the numerical tests presented here are also available on the wiki page of the code (https://github.com/demichie/IMEX_ SfloW2D/wiki), together with animations of the output and scripts to reproduce the results.

## 4.1 One dimensional test with discontinuous initial solution and topography

The first example is a 1D test for a Riemann problem with a discontinuous topography, as presented in Andrianov (2004)
and Kurganov and Petrova (2007). No frictional forces are considered in this test, aimed only at showing the capability of the numerical scheme used for the spatial discretization to properly model the propagation of strong discontinuities in both flow thickness and velocity.

The domain is the interval $[0;1]$ and bottom topography $B$ is the following step function:

$$
B(x) = \left\{
\begin{array}{ll}
2, & x \leq 0.5, \\
0.1, & x > 0.5.
\end{array}
\right.
$$

The gravitational constant is $g = 2$ and the initial data are:

$$
(w(x,o), u(x,0)) = \left\{
\begin{array}{ll}
(2.222, -1), & x \leq 0.5, \\
(0.8246, -1.6359), & x > 0.5.
\end{array}
\right.
$$

The initial solution (Figure 2, top panels) presents a discontinuity at $x = 0.5$ and the exact solution at $t > 0$ consists of a left-going rarefaction wave and a right-going shock-wave. We present here the numerical solution obtained with a 3 step
IMEX Runge-Kutta scheme, where the reconstruction from the cell centers to the faces is applied to the physical variables. The solutions obtained discretizing the domain with 100 cells (Figure 2, dashed red lines) and 400 cells (Figure 2, solid green lines) are compared. The numerical results at two different times $t > 0$ (Figure 2, middle and bottom panels) show the capacity of the numerical scheme to properly model the propagation of both the rarefaction and shock waves, and the good description of the shock with a small number of cells highlight the low numerical diffusivity of the central-upwind finite volume numerical
scheme implemented for the spatial discretization of the governing equations.

## 4.2 One dimensional problem with dry/wet interface and friction

This example is a 1D test for a system with friction, as presented in Kurganov and Petrova 2007. As in the previous test, an initial discontinuity is present, but this time representing the interface from a "wet" and a "dry" region. Thus, there is an additional numerical difficulty involving the capability of the numerical solver to propagate this discontinuities without creating regions with negative flow thickness. For this problem the bottom topography presents both smooth regions and a step, and it



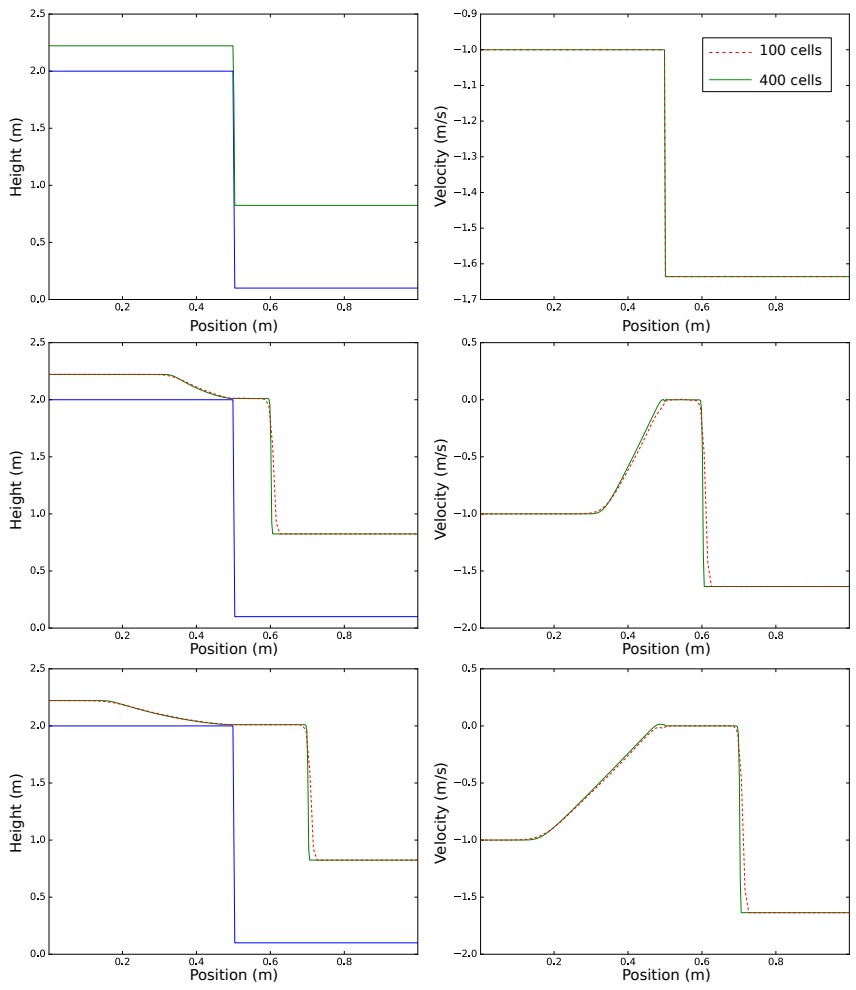

**Figure 2.** 1D Riemann problem with no friction. Numerical solution at three different times: 0s (top panels), 0.1s (middle panels), 0.2s (bottom panels). Flow thickness is plotted on the left panels, while the velocity is plotted on the right panels. The different colors represent numerical solutions obtained with different number of cells: 100 (dashed red line) and 400 (solid green line).

is defined as follows:

$$
B(x) = \begin{cases}
1, & x \leq 0, \\
\cos^2(\pi x), & 0 \leq x < 0.4, \\
\cos^2(\pi x) + 0.25(\cos(10\pi(x-0.5))+1), & 0.4 \leq x < 0.5, \\
0.5\cos^4(\pi x) + 0.25(\cos(10\pi(x-0.5))+1), & 0.5 \leq x < 0.6, \\
0.5\cos^4(\pi x), & 0.6 \leq x < 1, \\
0.25\sin(2\pi(x-1)), & 1 \leq x < 1.5, \\
0, & x \geq 1.5.
\end{cases}
$$



The initial conditions are defined by:

$$w(x,0) = \begin{cases} 1.4, & x \leq 0, \\ B(x), & x > 0, \end{cases} \qquad u(x,0) = 0.$$

For this simulation, the gravitational constant has value $g = 1$ and the friction coefficient is $\kappa = 0.001(1 + 10h)^{-1}$. Also for this test the numerical solution is obtained with a 3 step IMEX Runge-Kutta scheme, where the reconstruction from the cell centers to the faces has been applied to the physical variables and the domain $[-0.25; 1.75]$ has been discretized with 400 cells. The boundary conditions are prescribed to model a closed domain, also to check that the total mass contained in the domain

is kept constant by the numerical discretization schemes. The numerical solution at four different times (t=0s, t=0.2s, t=2s and t=40s) is presented in Fig. 3, showing the capability of the model to deal with the propagation of dry/wet interfaces and to reach a steady solution where the horizontal gradient of $w = h + B$ is null.

### 4.3 One dimensional pyroclastic avalanche with Voellmy-Salm friction

This example is a 1D test for the Voellmy-Salm rheology, with a pile of material initially at rest released on a constant slope

topography. This test is aimed at checking if the model is able to both preserve an initial steady condition, when the tangent of the pile free surface slope is smaller than the Coulomb friction coefficient, $\mu$, and to properly simulate the stopping of the flow, i.e. when inertial and gravitational forces are smaller than total basal friction,.

For this test, the domain is the interval $[0; 500]$ and the the center of the pile coincides with the center of the domain. The gravitational constant is $g = 9.81$ and the parameter of the Voellmy-Salm rheology are $\mu = 0.3$ and $\xi = 300$.

We present the results for a numerical simulation with a topography with a constant slope of 13° and an initial pile of material with a relative slope (i.e. with respect to the topography) of 20°. Thus, for this test, the initial condition is not steady and the pile of material start to move along the slope, until it reach the stoppage condition.

The numerical solution at three different times (t=0s, t=5s and t=40s) is presented in Fig. 4. For this simulation, the reconstruction technique with limiters has been applied to the physical variables and an IMEX 4-steps Runge-Kutta has been

adopted, while the domain has been discretized with 400 cells. The plot of the numerical solution at the intermediate time clearly shows that the front of the flow has a larger propagation velocity than the rest of the flow. On the other hand, comparing the left and right middle panels, we observe that part of the tail is not moving at all. After a few tens of seconds from the release, the flow reach a steady condition, as shown by the plot of the velocity at 40s (bottom-right panel in Fig. 4). This test highlights the capability of the model to reach a steady condition not only when the gradient of $w = h + B$ is null, as shown in

the previous example, but also with a flow with a positive slope below a critical condition.

### 4.4 Two dimensional pyroclastic avalanche with Voellmy-Salm friction

This test extends the simulation with a Voellmy-Salm rheological model presented in the previous section from one to two dimensions, with an example of an avalanche of finite granular mass sliding down an inclined plane merging continuously into a horizontal one. The initial conditions and the topography of this tutorial are the same as in Example 4.1 from Wang et al.





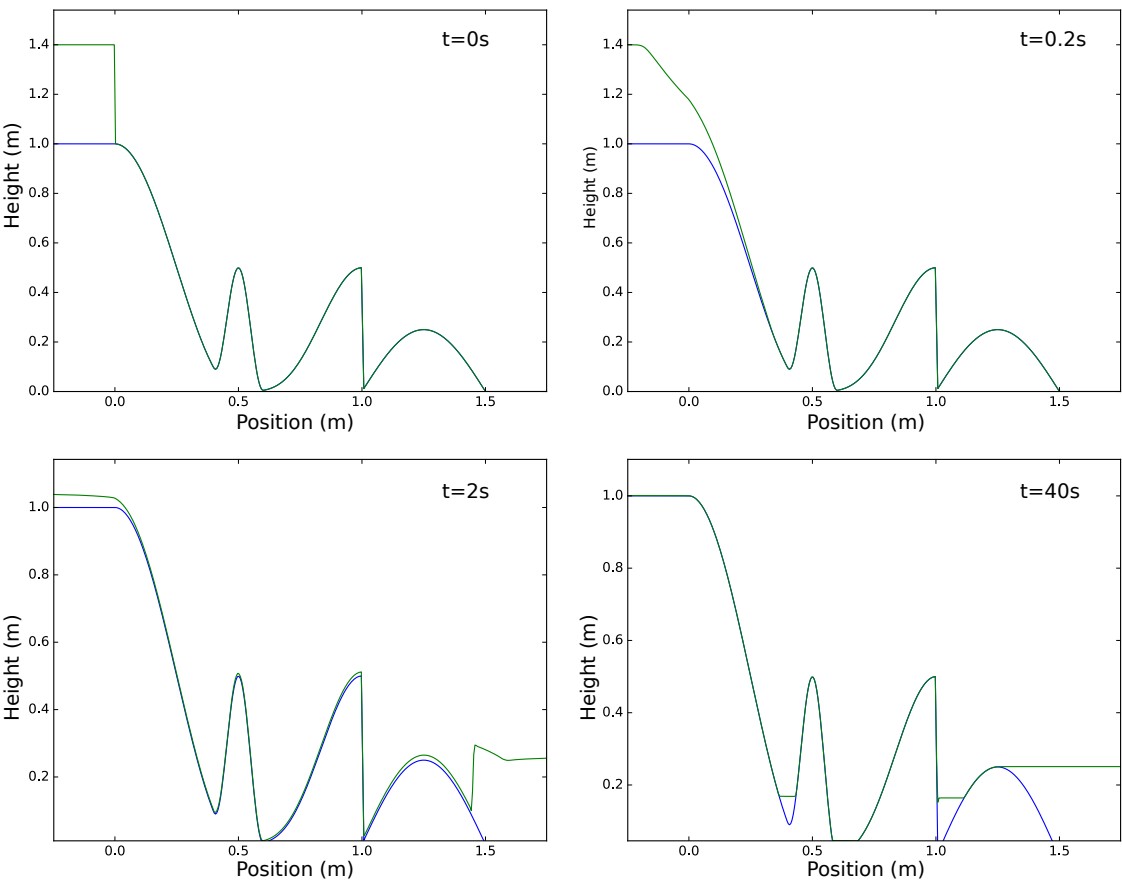

**Figure 3.** Numerical solution of a 1D Riemann problem with friction at four different times. The solid blue line represents the bathymetry, while the solid green line represents the free surface of the wet region.

(2004). The computational domain is the rectangle $[0; 30] \times [-7; 7]$, where an hemispherical shell holding the material together is suddenly released so that the bulk material starts to slide on an inclined flat plane at $35°$ (for $0 \leq x \leq 17.5$) into a horizontal run-out plane (for $x \geq 21.5$) connected by a smooth transition. Here we do not use the same rheological model as in the original paper of the example, but a Voellmy-Salm rhology is applied with $\mu = 0.3$ and $\xi = 300$.

5   The numerical solution at four different times (t=0s, t=7.5s, t=20s and t=25s) is presented in Fig. 5, where both the three-dimensional flow shape over the topography and thickness contours are presented. For this simulation, the reconstruction technique with limiters has been applied to the physical variables and an IMEX 2-steps Runge-Kutta has been adopted, while the domain has been discretized with $150 \times 100$ cells.

As shown by the plots presented in Fig. 5, the model is able to simulate the propagation of the flow with no numerical

10   oscillations or instabilities, without the need of artificial numerical diffusion. As the front reach the maximum runout and



**Figure 4.** Numerical solution of a 1D problem with Voellmy-Salm friction at three different times. In the left panels the topography (blue line) and the profile of the material (green line) are plotted. In the right panels the corresponding velocities are shown.

horizontal spreading (top-left panel), the tail of the flow is still accelerating and the avalanche body start to contract. Comparing the two panels at the bottom, it is evident that after about 20s the flow has stopped to propagate, with the deposit located at the transition region between the inclined and horizontal zones. This simulation took 238s on an Intel Corei5-3210M CPU at 2.50GHz.

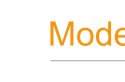
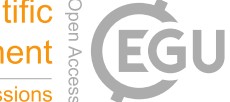


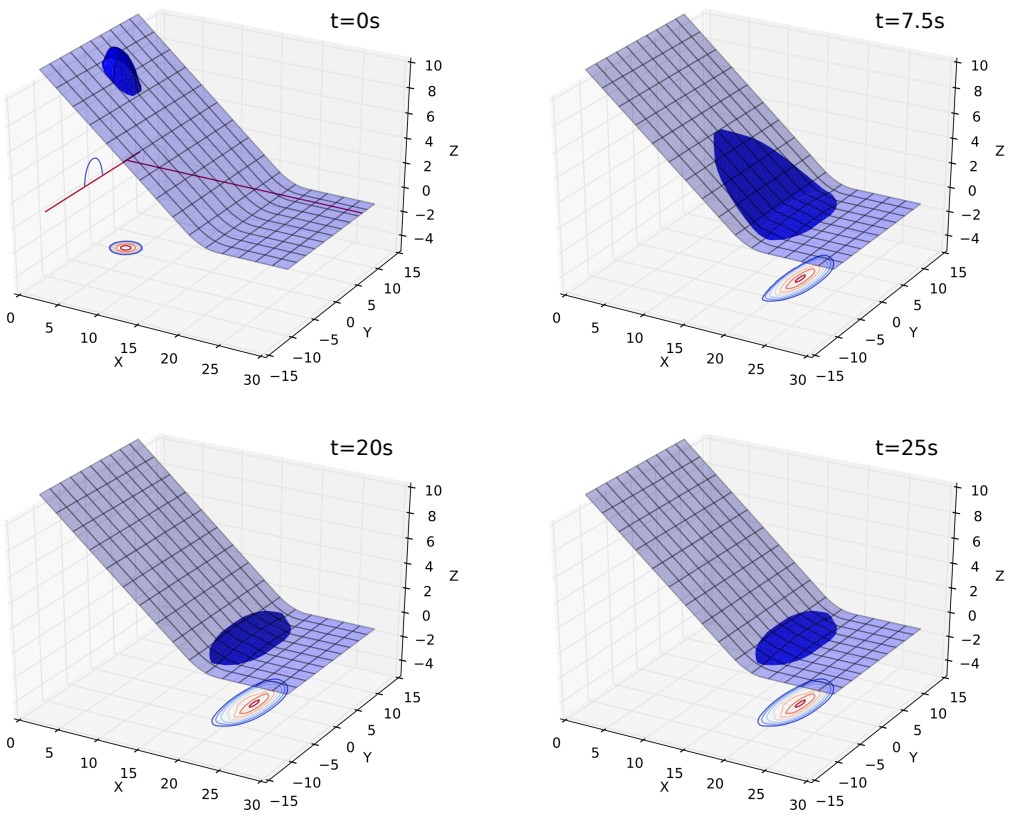

**Figure 5.** Numerical simulation of two dimensional pyroclastic avalanche with Voellmy-Salm friction. The contour plots on the bottom plane of each panel represent constant values of the thickness of the flow, whose free surface is represented in blue. A visual comparison between the two bottom plots highlight that a steady condition has been reached.

## 5 Simulation of a pyroclastic avalanche at Etna volcano

On February 11th, 2014, a hot, pyroclastic avalanche was generated at the New South-East Crater (NSEC) of Etna, triggered by the instability and collapse of its Eastern flank, where several vents were actively effusing lava flows towards Valle del Bove since January 22th. The avalanche propagation was recorded by INGV (Istituto Nazionale di Geofisica e Vulcanologia) monitoring system and, in particular, it was filmed by the thermal IR camera from Monte Cagliato, located on the east slope of the Valle del Bove at about 7 km from the NSEC, and Catania CUAD Visible Camera (ECV), at about 26 km south of the summit. A 500 m wide avalanche front propagated about 2.3 km along the steep slopes of the Valle del Bove before stopping at the break-in-slope, at the valley bottom. At the same time, a voluminous buoyant ash cloud was generated and rapidly dispersed in the North-North-East direction by an intense wind. The event is accurately reported by Andronico et al. (2018).





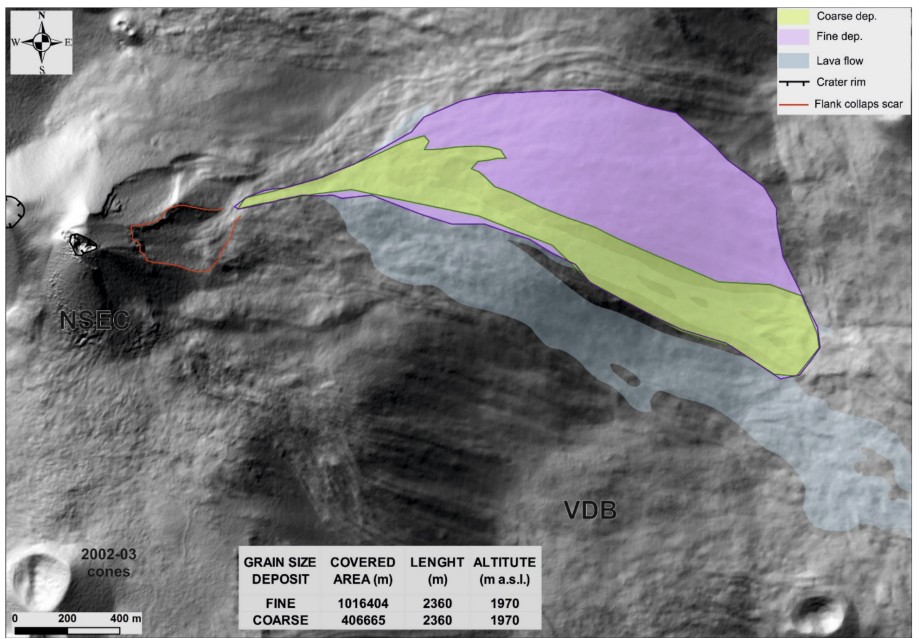

**Figure 6.** Map of the avalanche deposit of the February 2014 pyroclastic avalanche and of the lava flow.

The flank collapse left a detachment niche that allowed to estimate the avalanche volume to range between $0.5$ and $1.0 \times 10^6$ m$^3$. Due to difficult weather and environmental conditions, the presence of active lava flows in the region and the persistence of the Strombolian activity, it was unfortunately not possible to obtain a detailed map of the deposit thickness. Comparison of numerical results will therefore be limited to the flow boundary and runout.

IMEX_SfloW2D numerical simulations have been performed over the 2014 Digital Elevation Model of Mount Etna (De Beni et al., 2015). We have developed a MATLAB tool to modify the original topography by excavating a detaching volume with an ellipsoidal shape oriented towards the local maximum slope and with a prescribed total volume. The initial avalanche volume is defined by the difference between the original and the modified topography. It is also worth remarking that the DEM did not account for the presence of the thick lava flow that had been emplaced in the days before the avalanche event (Figure 6), which

likely controlled the avalanche path by confining it along its Southern edge (Andronico et al., 2018).

The avalanche rheological parameters have been varied in ranges consistent with previous studies of geophysical granular avalanches (e.g., Bartelt et al., 1999). In particular, $0.2 < \mu < 0.5$ and $300 < \xi < 5000$. Results of computations are reported graphically in Figure 7. The misfit between the simulated and observed flow path is due to the presence of the lava flow depicted in Figure 6 that was not included in the DEM. Overall, the fit is reasonable in terms of maximum runout and extension of the

deposit for runs d) ($\mu = 0.3, \xi = 500$) and f) ($\mu = 0.4, \xi = 5000$).

A thorough discussion about the optimal choice of the rheological model and parameters for pyroclastic avalanches would require an extensive comparison with similar phenomena occurred at Etna (few of which have been documented so far) and at

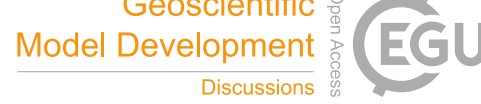

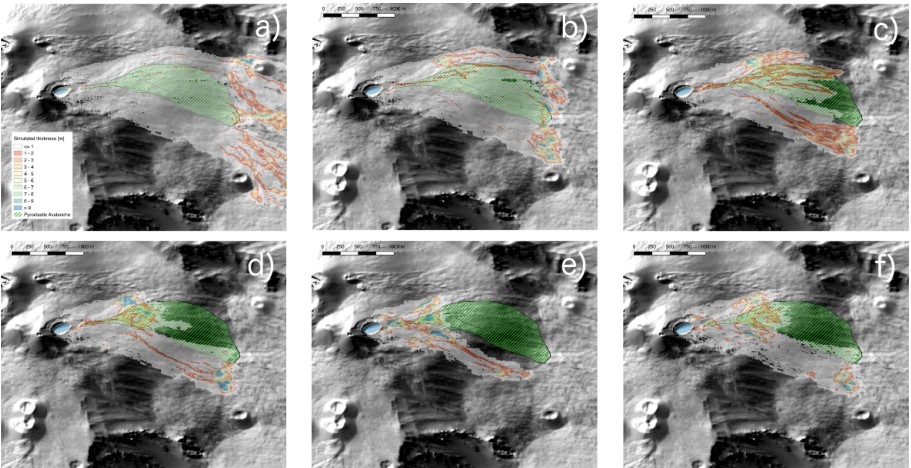

**Figure 7.** Results of IMEX_SfloW2D numerical simulations of a pyroclastic avalanche, overlapped to the hill-shaded relief of the Etna summit and the boundaries of the February 11th, 2014 event. Numerical parameters as follows: a) $\mu = 0.1$, $\xi = 500$; b) $\mu = 0.2$, $\xi = 500$; c) $\mu = 0.2$, $\xi = 100$; d) $\mu = 0.3$, $\xi = 500$; e) $\mu = 0.4$, $\xi = 500$; f) $\mu = 0.4$, $\xi = 5000$. Avalanche volume is equal to $0.5 \times 10^6$ m$^3$.

other analogous volcanoes, where more accurate measurements will be needed in the future to achieve a better calibration of the model. This is however clearly beyond the scope of the present work.

## 6   Conclusions

We have presented the physical formulation, numerical solution strategy and verification tests of the new IMEX_SfloW2D
numerical model for shallow granular avalanches. The numerical code is available open-source and freely downloadable from a GIT repository, where the users can also find the documentation and example tests described in this paper. The main features of the new model make it suited for the research and application to geophysical granular avalanches. In particular:

- The flexible discretization and numerical solution algorithm (not tied to the knowledge of the eigenstructure of the system of equations) allows to easily implement new transport equations.

- The formulation in Cartesian geographical coordinates is suited for running on Digital Surface Models (read in standard ESRI ASCII grid format) and for integration of non-hydrostatic terms, even on steep slopes.

- The conservative and positivity preserving numerical scheme allows a robust and accurate tracking of 1D and 2D discontinuities, including wet-dry interfaces and flow fronts.

- The implicit coupling of non-linear rheology terms allows the simulation of steady state equilibrium solutions and, in
particular, favours the flow stopping without the need of any ad-hoc empirical criterion.





   – The numerical procedure to evaluate the Jacobian of the non-linear system (based on a complex step derivative technique) allows an easy implementation and testing of new rheological models for complex geophysical granular avalanches.

*Code availability.* The numerical code, benchmark tests and documentation are available at https://github.com/demichie/IMEX_SfloW2D. Pre-processing scripts (to change the grid resolution and the numerical schemes) and post-processing scripts (to plot the solution variables and

5  to create animations) are also available. Furthermore, each example has a page description on the model Wiki (github.com/demichie/IMEX_SfloW2D/wiki) where detailed informations on how to run the simulations are given.

*Author contributions.* MdMV has developed the numerical algorithm and implemented the 1D model and numerical tests. TEO has contributed to the model formulation and application in the context of volcanological applications. GL has implemented the numerical scheme and algorithm in 2D. AA has tested the model and helped with comparison with similar depth-averaged models in volcanology.

10  *Competing interests.* No competing interests are present.

*Acknowledgements.* The work has been supported by the Italian Department of Civil Protection, INGV-DPC agreement B2 2016, Task D1. We warmly thank D. Andronico, E. De Beni and B. Behncke for useful discussions about the Etna 2014 event, and for providing field data and the Digital Elevation Model. TEO would like to thank P. Bartelt and B. Sovilla (SLF, Switzerland) for stimulating discussions on granular avalanches and density currents.



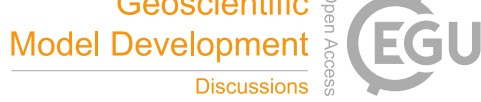

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
