# Peer review of "IMEX\_SfloW2D 1.0. A depth-averaged numerical flow model for pyroclastic avalanches"

_Geoscientific Model Development, 2018_

## Referee Comment (RC1) · O. Roche (Referee) · 9 Nov 2018

Review of "IMEX_SfloW2D 1.0. A depth-averaged numerical flow model for pyroclastic avalanches" by de' Michieli Vitturi et al., submitted to GMD.

Reviewer: Olivier Roche

General comments

This paper presents a new depth-averaged flow model for granular mixtures, with application to geophysical flows. The authors consider in particular pyroclastic avalanches generated during volcanic eruptions. This is a timely contribution because the vol-

canological community needs models to simulate dense granular flows. The paper presents the basic assumptions and equations used in the model as well as some validation tests, and as such represents a first step towards more detailed studies in specific volcanic contexts. Given the team's recognized international reputation in modeling of volcanic flows and the success of their previous studies, I have no doubt that the present model will lead to new fundamental understanding of granular flows and will be widely considered by the volcanological community. The manuscript is very well written and organized, and I recommend that the paper be accepted after minor revision. Please find below my general and specific comments.

The depth-averaged approach and the related equations as well as the discretization methods are well explained and justified. The authors discuss in particular how the case of zero flow thickness is treated. The main strengths of the model are the consideration of the effect of topography, which is critical for the simulation of geophysical flows on irregular topographies, the use of Digital Elevation Models, and the possible implementation of different granular flow rheologies. For their first tests and applications the authors have chosen a Voellmy-Salm rheology. This is a sensible choice as this rheology is often used to simulate geophysical flows such as snow avalanches for instance, and also because it involves both velocity independent dry granular friction and velocity dependent turbulent friction, which represent the respective contributions of granular friction and collisions, two mechanisms that operate in granular flows.

Specific comments

In introduction, I appreciate to effort made by the authors to distinguish between "avalanches" and "pyroclastic density currents" (PDCs), but the terms and the related natural phenomena could be introduced in a different way. In fact, the model addresses dense granular flows in general (irrespective of their volume), which are often present at the base of many PDCs, behave as described by the authors in lines 15-17 in page 2 for instance even though they are overridden by a dilute ash cloud (as considered actually in the test simulation presented in section 5), and may form deposits with low

aspect ratios. The model presented in this paper as more potential than the present introduction suggests. I think using the term "avalanche" is fine, but the authors could state that their model is applicable to dense pyroclastic avalanches (or "flows", or "currents") in general, which are basically concentrated granular flows (with negligible pore pressure in the present case).

In introduction, including a short discussion on the main similitudes and differences with earlier works would be certainly helpful. This would help the reader to appreciate for instance the choice of the spatial and temporal discretization schemes considered here.

The tests and simulations are important parts of this paper. Some complementary information would help to appreciate better some specific issues: - In section 4.2, the terms "wet" and "dry" should be defined clearly to avoid any possible confusion. I am not sure I understand the significance of these terms. - In section 5 on the simulation of pyroclastic avalanche at Etna volcano the authors should discussed, even briefly, the values of the parameters mu and xi they use. Though most readers will certainly appreciate that values of mu=0.2-0.5 are typical of most granular flows, values of xi=300-5000 may be more enigmatic. These values of xi are based on earlier works, but what are their significance in terms of physical processes? - It appears that the presence of a lava flow, not taken into account in the digital elevation model, probably influenced the emplacement of the pyroclastic avalanche. Could the authors run complementary simulations with a DEM including the lava flow, if available?

Page 1 L1-2. Pyroclastic avalanches generated from dome collapse should be mentioned as well. L4. 1 vol. % is a fairly low concentration for granular flows. Say rather 10-50 vol. %? L11. The term "wet-dry" should be defined. L21. Debris flows are water-saturated, which is not the case of the other flows mentioned here. L24. I appreciate this point! I think the term "avalanche" is fine.

Page 2 L4-5. Please see my general comments on application of the model for pyroclastic density currents. L22. There is no section 1.2. Delete section 1.1?

Page 3 L7. Typo (conservative) L28-29. This is certainly of major strength of this new code.

Page 4 L6-8. This sentence is long and not clear. Are there words mission? It could be said also that the St-Venant approach is relevant when the flow depth is significantly smaller than the flow length.

Page 5 L10. To model shallow pyroclastic avalanches (?) L13. Is the term "fits" appropriate? Say rather "relevant"? L20-21. It could be stated that the velocity dependent turbulent friction is commonly considered to correspond to granular collisions.

Page 6. L31. This suggests. . . Page 8. L5. What do you mean by "velocities too large"? Please clarify. Page 9. L12. . . .because such scheme is well-suited. . . (?) Page 10. L9-10. This is an important issue. Is it similar to or different than other models involving a Voellmy-Salm rheology?

Page 11. L4. Andrianov (2004) is not in the list of references. L8-15. In Figure 2, what is the blue line? I guess this is not the initial flow thickness otherwise the initial model solutions would not correspond to this boundary condition. Is it the initial topography as in Figs. 3 and 4? L18. The terms "wet" and "dry" must be defined here.

Page13. L23. . . .the flow reaches. . . Page14. L10. . . .the front reaches. . . Page15. In Fig. 5 at t=7.5 s it seems that the flow thickness represented on the bottom plane is zero at x<17-18 whereas some material is present on the inclined plane. Is it because the flow thickness on the bottom plane is not represented below a threshold value, or else? Page16. L8-9. This sentence may suggest another event. State that the ash cloud was generated by the avalanche.

Page 17. L11-12. Please see my general comments on the rheological parameters. L14-15. The term "extension" may be ambiguous. In fact, the fit is reasonable in terms of the area covered by the model deposit, which is close to that of the pyroclastic

avalanche though shifted toward the south because of the absence of the lava flow in the model (please see also my general comments). It could be stated as well that values of mu=0.3-0.4 correspond to fairly low friction coefficients of dry granular materials, which is an interesting result. L16 and next page. I subscribe to this point of view.
* * *

---

## Referee Comment (RC2) · K. Kelfoun (Referee) · 20 Nov 2018

This manuscript presents the mathematical basis of a new depth-averaged flow model for geophysical flows as well as some simulations that show the capabilities and the accuracy of the code.

The two main strengths of the new model are: 1) that the equations are written in global Cartesian coordinates, with the z-axis parallel to the gravity. 2) that the source code is available for the community.

The manuscript is clearly written and well organized. To me, from both a modelling and

a volcanological points of view, this is a good article that merits to be published after minor revisions.

I have read the review of RC1 and I agree with nearly all his comments. I won't repeat all the points he has raised and I just focus on the most significant points and additional comments.

1) Avalanches, PDC and the field case used :

I recognize that there is no real consensus in the volcanological community to name the different "currents" that are observed: pyroclastic flows, avalanches, surges, etc. However, I think that definitions of the paper will add to the confusion. PDCs is a general term that was introduced to include all the "currents". It only means that the "currents" move because their densities are higher than their environment, whatever their physics. In this definition, an avalanche is a density current. The authors can use the term "avalanche", but they do not need to redefine (nor to use) the term PDCs.

I also want to point out a related source of potential confusion in the manuscript. During my first reading, even if the authors use the term "avalanche", I thought that the subject was the simulation of "pyroclastic flows". I was thus surprised that the dome collapses were forgotten as a source of the "avalanches" in the abstract. The difference between "pyroclastic flows" and "pyroclastic avalanches" is in the runout: the runout of a "pyro-clastic avalanche" can be simulated as a dry granular material. Additional phenomena occur in the "pyroclastic flows" that gives to them the very high fluidity observed in the field. I agree with the authors: the field case used (section 5) is a pyroclastic avalanche.

If the authors want to focus on pyroclastic avalanche only, they need to explain more clearly the difference with the "pyroclastic flows" in the introduction. However, I think that the authors can extend the application domain of their model to pyroclastic flows (in this case, they must include dome collapses and vulcanian explosions in the descrip-tion of their origins). In both cases, to avoid future misinterpretations of the best values they have obtained at Etna (mu = 0.3-0.4) in the debate of the fluidity of pyroclastic

flows, it is important that they recall the "avalanche" nature of the natural phenomenon simulated in section 5 and that they describe at least briefly the topography characteristics: runout (2300 m) but also elevation differences ($\sim$1200 m) and mean slope ($\sim$25°). Page 16, line 8, for example.

2) Equations:

To me, the equations are clearly presented and seem to be correct. The tests are relevant and convincing. However, the manuscript is more a mathematical paper with strong implications in volcanology rather than a volcanological paper. Because RC1 and I are not mathematicians, the opinion of a specialist of these systems of equations and of their numerical resolution is required.

Three points are not clear for me and must be explicitly explained in the text: - Where the valleys turn rapidly (in an almost horizontal plan), do the system of equations presented take into account the centrifugal force that will increase the apparent gravity and then the retarding stress? - Is the modular structure of the code (Page 1, line 16) compatible with a rheology that does not include a term related to the flow velocity if we want to model a purely frictional or plastic flow, for example? Or do the user need to add an artificial viscosity (or turbulence) and, in this case, how can be determined the lowest viscosity required? - It is possible to use the full resolution of a topography or does the initial interpolation smooth it (lines 9-10, page 6)? The resolution has a strong influence for flow simulations in narrow valleys.

Other minor comments:

- Page 1, line 25: better to use the past: 'the French avaler, which meant "move down the valley" '. Today it has been replaced by "dévaler" and the meaning of "avaler" has evolved and it is used for "swallow" (move down but to the stomach).

- Page 2, line 17: category

- Page 3, line 7: conservative

- Page 3, line 14: perhaps you could explain the origin of the name of your code.

- Page 6, line 12-14: I do not understand this sentence. Could you explain it a little further?

- Page 13, line 16: the friction coefficient is not related to a Voellmy-Salm rheology. For clarity, please indicate that you are using another rheology, that of Kurganov and Petrova (2017). A similar indication – that the values come from already published examples - can introduce the "strange" values used in the equations of line 9 (why 2.2222, 0.8246, -1.6359, etc.?)

- A table of all the variables used and their meaning would be useful.

---

## Author Comment (AC1) · 16 Jan 2019

[]letter

Dear Editor,

please find in the next pages the point-by-point responses to the reviewers comments on the manuscript entitled "IMEX_SfloW2D 1.0. A depth-averaged numerical flow model for pyroclastic avalanches". We found the comments very interesting and helpful to improve the quality of the work. In addition, as supplement to this comment, ha have added a pdf of the manuscript where we have highligthed the changes done.

[Figure]

Best Regards,

Mattia de' Michieli Vitturi (*) and co-authors

]

* Corresponding Author

[Figure]

**C.** *Page 3 L1-5. ... the authors could state that their model is applicable to dense pyroclastic avalanches (or "flows", or "currents") in general.*

**R.** The model presentation with respect to the natural phenomenon has been rewritten in the introductory paragraph, also in response to Reviewer 2 comments. A sentence has been added at lines 26–29 to clarify the needed improvements that would make the model suitable for the simulation of more general (and voluminous) pyroclastic flows.

**C.** *Page 3 L6-7. In introduction, including a short discussion on the main similitudes and differences with earlier works would be certainly helpful.*

**R.** A comparative discussion with earlier works is presented in Section 1.1. for what concerns the modelling aspects. We think that this was a pertinent choice, since the paper (and the journal) is more focused on the modelling aspects than on the volcanological applications. However, to respond to the reviewer's comment, we have further extended the introductory part (at lines 12–17) describing previous applications in more detail.

**C.** *Page 3 L1-5. In section 4.2, the terms "wet" and "dry" should be defined clearly to avoid any possible confusion.*

**R.** We defined the terms at the beginning of Section 4.2. *"As in the previous test, an initial discontinuity is present, but this time representing the interface from a "wet" (presence of flow) and a "dry" (no flow or zero thickness) region, with the terminology borrowed from the common use of Saint-Venant equations in hydrology".*

**C.** *Page 3 L15. In section 5 on the simulation of pyroclastic avalanche at Etna volcano the authors should discussed, even briefly, the values of the parameters mu and xi they use. Though most readers will certainly appreciate that values of mu=0.2-0.5 are typical of most granular flows, values of xi=300- 5000 may be more enigmatic. These values of xi are based on earlier works, but what are their significance in terms of*

*physical processes?*

**R.** We have addedd a sentence in Sect. 5 For quasi-static granular flows, $\mu$ is physically related to the basal friction of the granular material. However, its value for rapid avalanches cannot be easily defined a priori. We thus simply consider $\mu$ and $\xi$ as empirical model parameters whose values need to be calibrated.

**C.** *Page 3 L20. Could the authors run complementary simulations with a DEM including the lava flow, if available?*

**R.** Unfortunately, the DEM including the 2014 lava flow is still not available to the authors.

**C.** *Page 1 L1-2. Pyroclastic avalanches generated from dome collapse should be mentioned as well.*

**R.** Dome collapse is now mentioned as a possible generating mechanism of pyroclastic avalanches (P2 L 24).

**C.** *P2 L4. 1 vol. % is a fairly low concentration for granular flows. Say rather 10-50 vol. %?*

**R.** We agree. This was a typographic error. It has been corrected.

**C.** *L21. Debris flows are water-saturated, which is not the case of the other flows mentioned here.*

**R.** We agree. Debris flows are no longer mentioned in the list.

**C.** *Page 3 L7. Typo (conservative)*

**R.** Fixed

**C.** *Page 4 L6-8. This sentence is long and not clear. Are there words mission? It could be said also that the St-Venant approach is relevant when the flow depth is significantly smaller than the flow length.*

[Figure]

**R.** The sentence has been rewritten in a simpler way. In addition, as suggested, the following text has been added: "*Saint Venant's equations are partial differential equations suitable when the flow horizontal length scale is much greater than the vertical one, allowing to disregard vertical flow motion.*"

**C.** *Page 5 L10. To model shallow pyroclastic avalanches (?)*

**R.** Now it reads: "*To properly model shallow pyroclastic avalanches...*".

**C.** *L13. Is the term "fits" appropriate? Say rather "relevant"?*

**R.** Fixed changing to "relevant".

**C.** *L20-21. It could be stated that the velocity dependent turbulent friction is commonly considered to correspond to granular collisions.*

**R.** This is now stated in the text.

**C.** *Page 6. L31. This suggests. . .*

**R.** Fixed

**C.** *Page 8. L5. What do you mean by "velocities too large"? Please clarify.*

**R.** This is explained in a preceding paragraph that has been extended: "*In this case, a correction procedure is required to limit the values of the velocity components at the interfaces when the flow thickness goes to zero, as done in* ?*"*. The main point here is that, from the centers of the cells, we reconstruct independently the values of the free surface ($w = h + B$) and of the momentum ($hu$) at the cell faces. Then, thickness is obtained from $h = w - B$ and flow velocity is calculated dividing the momentum by the thickness. When the linear extrapolation of thickness goes to 0 at the face, while the absolute value of the momentum remains larger than 0, the division results can grow to unrealistic values of the velocity. For this reason, we introduced in the model the possibility to do a linear reconstruction of the physical variables $\mathbf{U} = (w, u, v)^T$, in order to avoid the aforementioned division.

**C.** *Page 9. L12. . . .because such scheme is well-suited. . . (?)*

**R.** Fixed

**C.** *Page 10. L9-10. This is an important issue. Is it similar to or different than other models involving a Voellmy-Salm rheology?*

**R.** Unfortunately, the numerical schemes adopted in other models is not always well-documented, so it is not possible for us to comment on other implementations of stopping conditions.

**C.** *Page 11. L4. Andrianov (2004) is not in the list of references.*

**R.** The reference has been added to the list.

**C.** *L8-15. In Figure 2, what is the blue line? I guess this is not the initial flow thickness otherwise the initial model solutions would not correspond to this boundary condition. Is it the initial topography as in Figs. 3 and 4?*

**R.** The blue line represent the topography, and this is now stated in the caption of the figure.

**C.** *L18. The terms "wet" and "dry" must be defined here.*

**R.** We defined the terms at the beginning of Section 4.2. *"As in the previous test, an initial discontinuity is present, but this time representing the interface from a "wet" (presence of flow) and a "dry" (no flow or zero thickness) region, with the terminology borrowed from the common use of Saint-Venant equations in hydrology".*

**C.** *Page13. L23. . . .the flow reaches. . .*

**R.** Fixed

**C.** *Page14. L10. . . .the front reaches. . .*

**R.** Fixed

**C.** *Page15. In Fig. 5 at t=7.5 s it seems that the flow thickness represented on the bottom plane is zero at x<17-18 whereas some material is present on the inclined plane. Is it because the flow thickness on the bottom plane is not represented below a threshold value, or else?*

**R.** This is now explained in the caption of the Figure. The outermost contour corresponds to a thickness of 0.06m, thus the thinner portion of the flow is not represented in the contour plot. For this test, also the maximum initial thickness (1.85m) was not reported in the text, so we added it.

**C.** *Page16. L8-9. This sentence may suggest another event. State that the ash cloud was generated by the avalanche.*

**R.** We have specified in the text (P18, L10) that "At the same time, a voluminous buoyant ash cloud was generated by elutriation of the finest ash from the avalanche".

**C.** *Page 17. L11-12. Please see my general comments on the rheological parameters.*

**R.** We have added a sentence in Sect. 5 For quasi-static granular flows, $\mu$ is physically related to the basal friction of the granular material. However, its value for rapid avalanches cannot be easily defined a priori. We thus simply consider $\mu$ and $\xi$ as empirical model parameters whose values need to be calibrated.

**C.** *L14-15. The term "extension" may be ambiguous. In fact, the fit is reasonable in terms of the area covered by the model deposit, which is close to that of the pyroclastic avalanche though shifted toward the south because of the absence of the lava flow in the model (please see also my general comments). It could be stated as well that values of mu=0.3-0.4 correspond to fairly low friction coefficients of dry granular materials, which is an interesting result.*

**R.** We have specified that we refer to the areal extension of the modelled deposit and added a comment (P18 L30) "It is interesting to notice that the value $\mu = 0.3 - 0.4$ corresponds to fairly low friction coefficients of dry granular materials (corresponding

to repose angles of $16° − 20°$). This is commonly acknowledged as frictional weakening for rapid granular flows."

**C.** *P2 L12. The authors can use the term "avalanche", but they do not need to redefine (nor to use) the term PDCs.*

**R.** We have not redefined the term Pyroclastic Density Current but we think that it is still useful to put it in the context of previous fluid dynamics literature in other fields.

**C.** *P2 L21. If the authors want to focus on pyroclastic avalanche only, they need to explain more clearly the difference with the "pyroclastic flows" in the introduction.*

**R.** We agree with the Reviewer and added a sentence introducing the classical "pyroclastic flow" definition (P1, lines 12–17) and the improvements needed to extend the applicability of the presented model to more general pyroclastic currents (P1, lines 26–29).

**C.** *Where the valleys turn rapidly (in an almost horizontal plan), do the system of equations presented take into account the centrifugal force that will increase the apparent gravity and then the retarding stress?*

**R.** The centrifugal force is usually added in depth-averaged models written in local, terrain-following coordinates. This is not considered in our model, and this is now stated in the manuscript (P. 5 line 8). For absolute, geographical coordinates, curvature and rapid topographic changes would result in non-hydrostatic acceleration terms, which are neglected in the present version of the model (P. 3 line 25)

**C.** *Is the modular structure of the code (Page 1, line 16) compatible with a rheology that does not include a term related to the flow velocity if we want to model a purely frictional or plastic flow, for example? Or do the user need to add an artificial viscosity (or turbulence) and, in this case, how can be determined the lowest viscosity required?*

**R.** The Coulomb friction term used in model and discussed in section 2.2 is already a velocity independent term. In section 3.2 it is discussed how this term is treated

numerically to enforce the stopping condition. No artificial viscosity is needed to treat these rheologies.

**C.** *It is possible to use the full resolution of a topography or does the initial interpolation smooth it (lines 9-10, page 6)? The resolution has a strong influence for flow simulations in narrow valleys.*

**R.** We agree with the reviewer that the resolution has a strong influence in narrow valleys. Conversely, we have checked that, in absence of narrow valleys, the resolution does not change significantly the runout of the simulations. In any case, we have clarified in the text that, when the DEM and the computational grid resolutions are the same, there is no interpolation and consequent degradation of the topography.

Other minor comments

**C.** *Page 1, line 25: better to use the past: 'the French avaler, which meant "move down the valley" '. Today it has been replaced by "dévaler" and the meaning of "avaler" has evolved and it is used for "swallow" (move down but to the stomach).*

**R.** We have specified in the introduction that the term derives from the old French.

**C.** *Page 2, line 17: category*

**R.** Fixed

**C.** *Page 3, line 7: conservative*

**R.** Fixed

**C.** *Page 3, line 14: perhaps you could explain the origin of the name of your code.*

**R.** The IMEX in the name of the code comes from the Implicit/Explicit solver we used, while SfloW stands from Shallow Water flow. This is now written in the manuscript.

**C.** *Page 6, line 12-14: I do not understand this sentence. Could you explain it a little*

[Figure]

*further?*

**R.** This is explained in details at the end of Section 3.1. We add here that one important steady state of the system is represented by the conditions $h + B = const$, $u = 0$. In equations 2-4, such conditions immediately lead to a steady state, but in the numerical model the last term appearing in equations 3 and 4, related to the gradient of the free surface $h + B$, is split in several terms of $\mathbf{F}$, $\mathbf{G}$ and $\mathbf{S_1}$, as defined by equation 6. Thus, the discretization of the different terms have to be consistent to preserve the steady state, and when a high-order scheme with a bi-linear reconstruction (from centers to faces) of the solution is adopted, also for the topography we need the face values to be bi-linear functions, thus with the center values corresponding to the average value of the face ones.

**C.** *Page 13, line 16: the friction coefficient is not related to a Voellmy-Salm rheology. For clarity, please indicate that you are using another rheology, that of Kurganov and Petrova (2017). A similar indication – that the values come from already published examples - can introduce the "strange" values used in the equations of line 9 (why 2.2222, 0.8246, -1.6359, etc.?)*

**R.** We have specified in the text that we are using a simpler friction term $(-\kappa(h)u)$. At the beginning of the description of this test, as for the previous one, we have written that the test is "*as presented in Kurganov and Petrova 2007*".

**C.** *A table of all the variables used and their meaning would be useful.*

**R.** We added a table to the manuscript.

**Supplement:**

[revised manuscript text omitted]